# DEEP POWER LAWS
# FOR HYPERPARAMETER OPTIMIZATION

## ABSTRACT

Hyperparameter optimization is an important subfield of machine learning that focuses on tuning the hyperparameters of a chosen algorithm to achieve peak performance. Recently, there has been a stream of methods that tackle the issue of hyperparameter optimization, however, most of the methods do not exploit the scaling law property of learning curves. In this work, we propose Deep Power Law (DPL), a neural network model conditioned to yield predictions that follow a power-law scaling pattern. Our model dynamically decides which configurations to pause and train incrementally by making use of multi-fidelity estimation. We compare our method against 7 state-of-the-art competitors on 3 benchmarks related to tabular, image, and NLP datasets covering 57 diverse search spaces. Our method achieves the best results across all benchmarks by obtaining the best any-time results compared to all competitors. We open-source our implementation and make our code publicly available at: https://anonymous.4open.science/r/DeepRegret-0F61/

## 1 INTRODUCTION

Hyperparameter Optimization (HPO) is a major challenge for the Machine Learning community. Unfortunately, HPO is not yet feasible for Deep Learning (DL) methods due to the high cost of evaluating multiple configurations. Recently, Gray-box HPO (a.k.a. multi-fidelity HPO) has emerged as a promising paradigm for HPO in DL, by discarding poorly-performing hyperparameter configurations after observing the validation error on the low-level fidelities of the optimization procedure (Li et al., 2017; Falkner et al., 2018; Awad et al., 2021; Li et al., 2020). The advantage of gray-box HPO compared to online HPO (Chen et al., 2017; Parker-Holder et al., 2020), or meta-gradient HPO (Maclaurin et al., 2015; Franceschi et al., 2017; Lorraine et al., 2020) is the ability to tune all types of hyperparameters.

In recent years, a stream of papers highlights the fact that the performance of DL methods is predictable (Hestness et al., 2017), concretely, that the validation error rate is a power law function of the model size, or dataset size (Rosenfeld et al., 2020; 2021). Such a power law relationship has been subsequently validated in the domain of NLP, too (Ghorbani et al., 2022). In this paper, we demonstrate that the power-law principle has the potential to be a game-changer in HPO, because we can evaluate hyperparameter configurations in low-budget regimes (e.g. after a few epochs), then estimate the performance on the full dataset using dataset-specific power law models.

We introduce Deep Power Law (DPL) ensembles, a probabilistic surrogate for Bayesian optimization (BO) that estimates the performance of a hyperparameter configuration at future budgets using ensembles of deep power law functions. Subsequently, a novel proposed flavor of BO dynamically decides which configurations to pause and train incrementally by relying on the performance estimations of the surrogate. We demonstrate that our method achieves the new state-of-the-art in HPO for DL by comparing against 8 strong HPO baselines, and 57 datasets of three diverse modalities (tabular, image, and natural language processing). As a result, we believe the proposed method has the potential to finally make HPO for DL a feasible reality. Overall, our contributions can be summarized as follows:

- We introduce a novel probabilistic surrogate for gray-box HPO based on ensembles of deep power law functions.

- We derive a simple mechanism to combine our surrogate with Bayesian optimization.
- Finally, we demonstrate the superiority of our method against the current state-of-the-art in HPO for Deep Learning, with a very large-scale HPO experimental protocol.

## 2 RELATED WORK

**Multi-fidelity HPO** relaxes the black box assumption by assuming it has access to the learning curve of a hyperparameter configuration. Such a learning curve is the function that maps either training time or dataset size, to the validation performance. The early performance of configurations (i.e. first segment of the learning curve) can be used to discard unpromising configurations, before waiting for full convergence. Successive halving (Jamieson & Talwalkar, 2016) is a widely used multi-fidelity method that randomly samples hyperparameter configurations, starts evaluating them, and ends a fraction of them upon reaching a predefined budget. Afterward, the budget is multiplied by the fraction of discarded hyperparameter configurations and the process continues until the maximum budget is reached. Although the method relies only on the last observed value of the learning curve, it is very efficient. In recent years, various flavors of successive halving have been elaborated, including Hyperband (Li et al., 2017), which effectively runs successive halving in parallel with different settings. A major improvement to Hyperband is replacing random search with a more efficient sampling strategy (Awad et al., 2021; Falkner et al., 2018). However, the only assumption these methods make about the learning curve is that it will improve over time. In contrast, we fit surrogates that exploit a power law assumption on the curves.

**Learning curve prediction** is a related topic, where the performance of a configuration is predicted based on a partially observed learning curve. Typically, the assumptions about the learning curve are much stronger than those described above. The prediction is often based on the assumption that the performance increases at the beginning and then flattened towards the end. One way to model this behavior is to define a weighted set of parametric functions (Domhan et al., 2015; Klein et al., 2017). Then, the parameters of all functions are determined so that the resulting prediction best matches the observed learning curve. Another approach is to use learning curves from already evaluated configurations and to find an affine transformation that leads to a well-matched learning curve (Chandrashekaran & Lane, 2017). A more data-driven approach is to learn the typical learning curve behavior directly from learning curves across different datasets (Wistuba & Pedapati, 2020). Learning curve prediction algorithms can be combined with successive halving (Baker et al., 2018). In contrast to this line of research, we actually fit ensembles of power law surrogates for conducting multi-fidelity HPO with Bayesian optimization.

**Scaling laws** describe the relationship between the performance of deep learning models as a function of dataset size or model size. Concretely, Hestness et al. (2017) show empirically for different data modalities and neural architectures that a power law relationship holds when growing the dataset. Further work confirms this observation and extends it by demonstrating the power law relationship also with regard to the model size (Rosenfeld et al., 2020; 2021; Ghorbani et al., 2022). From a practical angle, Yang et al. (2022) propose to tune hyperparameters on a small-scale model and then transfer it to a large-scale version. In contrast to these papers, we directly use the power law assumption for training surrogates in Bayesian optimization for HPO.

## 3 PRELIMINARIES

**Hyperparameter Optimization (HPO)** demands finding the configurations $\lambda \in \Lambda$ of a Machine Learning method that achieve the lowest validation loss $\mathcal{L}^{(\text{Val})}$ of a model (e.g. a neural network), which is parameterized with $\theta$ and learned to minimize the training loss $\mathcal{L}^{(\text{Train})}$ as:

$$\lambda^* := \underset{\lambda \in \Lambda}{\arg\min} \ \mathcal{L}^{(Val)}\left(\lambda, \theta^*\left(\lambda\right)\right),$$

$$\text{s.t.} \quad \theta^*\left(\lambda\right) := \underset{\theta \in \Theta}{\arg\min} \ \mathcal{L}^{(Train)}\left(\lambda, \theta\right) \tag{1}$$

For simplicity we denote the validation loss as our function of interest $f(\lambda) = \mathcal{L}^{(Val)}\left(\lambda, \theta^*\left(\lambda\right)\right)$. The optimal hyperparameter configurations $\lambda^*$ of Equation 1 are found via **an HPO policy** $\mathcal{A}$ (also

called an HPO method) that given a history of $N$ evaluated configurations $H := \{\lambda_i, f(\lambda_i)\}_{i=1}^{N}$ suggests the $(N+1)$-th configuration to evaluate as $\lambda_{N+1} := \mathcal{A}(H)$ where $A : [\Lambda \times \mathbb{R}_+]^N \to \Lambda$. The search for an optimal HPO policy is a bi-objective problem in itself, aiming at (i) finding a configuration out of $N$ evaluations that achieves the smallest validation loss $f(\lambda)$, and (ii) ensuring that the costs of evaluating the $N$ configurations do not exceed a total budget $\Omega$, as shown in Equation 2.

$$\arg\min_{\mathcal{A}} \min_{i \in \{1, \dots, N\}} f\left(\lambda_i = \mathcal{A}\left(H^{(i-1)}\right)\right), \tag{2}$$

$$\text{where:} \quad H^{(i)} := \begin{cases} \{(\lambda_j, f(\lambda_j))\}_{j=1}^{i} & i > 0 \\ \emptyset & i = 0 \end{cases}$$

$$\text{subject to:} \quad \Omega > \sum_{i=1}^{N} \text{cost}\left(f(\lambda_i)\right)$$

**Bayesian optimization (BO)** is the most popular type of policy for HPO, due to its ability to balance the exploration and exploitation aspects of minimizing the loss $f$. Technically speaking, BO fits a surrogate $\hat{f}(\lambda; \theta)$ parametrized with $\theta$ to approximate the observed loss $f(\lambda)$ using the history $H$, as $\theta^* := \arg\min_\theta \mathbb{E}_{(\lambda, f(\lambda)) \sim p_H} p(f(\lambda) | \lambda, \theta)$. Afterwards, BO uses an acquisition/utility function $a : \Lambda \to \mathbb{R}_+$ to recommend the next configuration as $\lambda_{N+1} := \mathcal{A}\left(H^{(N)}\right) = \arg\max_{\lambda \in \Lambda} a(\lambda; \theta^*)$. A typical acquisition choice is the Expected Improvement (Mockus et al., 1978). For a more detailed introduction to BO and HPO we refer the interested reader to Hutter et al. (2019)

**Gray-box (multi-fidelity) HPO** refers to the case where an approximation of the validation loss can be measured at a lower budget $b \in B$, where $B = (0, \text{max\_budget}]$. For instance in Deep Learning we can measure the validation loss after few epochs ($0 < b < \epsilon$), rather than wait for a full convergence ($b = \text{max\_budget}$). Throughout this paper the term budget refers to the number of optimization epochs. The evaluation of a configuration $\lambda$ for a budget $b$ is defined as $f(\lambda, b) := \mathcal{L}^{(Val)}(\lambda, \theta^*(\lambda, b))$, where $f(\lambda, b) : \Lambda \times B \to \mathbb{R}_+$. The concept of budgets alters the HPO problem definition slightly. The history of $N$ configurations evaluated at different budgets becomes a set of $N$ triples (config, budget, eval) defined as $H := \{(\lambda_i, b_i, f(\lambda_i, b_i))\}_{i=1}^{N}$. A gray-box HPO policy is still optimized for Equation 2, however, the constraint is altered as $\Omega > \sum_{i=1}^{N} \text{cost}(f(\lambda_i, b_i))$.

## 4 POWER LAW SURROGATES FOR BAYESIAN OPTIMIZATION

Prior work has demonstrated that the performance of Machine Learning methods as a function of budgets (i.e. dataset size, number of optimization epochs, model size, image resolution) follows a power law relationship (Rosenfeld et al., 2020; 2021). In this work, we employ this power law dependence between the validation loss and the number of optimization epochs in Deep Learning. We propose a novel gray-box Hyperparameter Optimization method which is based on power law surrogates. We assume that every learning curve $f(\lambda, \cdot)$ can be described by a power law function defined by $(\alpha, \beta, \gamma)$. Concretely, we define a power law function modelling the validation loss for a configuration $\lambda$ at budget $b$ (a.k.a. number of epochs) as shown in Equation 3.

$$\hat{f}(\lambda, b) := \alpha_\lambda + \beta_\lambda \, b^{-\gamma_\lambda}, \quad \alpha_\lambda, \beta_\lambda, \gamma_\lambda \in \mathbb{R} \tag{3}$$

We propose to use surrogate models employing the power law in Bayesian optimization (BO) to optimize hyperparameters. We suggest learning a **shared power law function** across all configurations by conditioning the power law coefficients $\alpha, \beta, \gamma$ on $\lambda$ using a parametric neural network $g$ that maps a configuration to the power law coefficients of its learning curve as $g : \Lambda \to \mathbb{R}^3$. The network $g$ has three output nodes, corresponding to the power law coefficients, denoted as $g(\lambda)_\alpha, g(\lambda)_\beta, g(\lambda)_\gamma$. The configuration-conditioned power law surrogate becomes:

$$\hat{f}(\lambda, b) := g(\lambda)_\alpha + g(\lambda)_\beta \, b^{-g(\lambda)_\gamma}, \quad g : \Lambda \to \mathbb{R}^3 \tag{4}$$

Using a history of learning curve evaluations $H := \{(\lambda_i, b_i, f(\lambda_i, b_i))\}_{i=1}^N$ we can train the power law surrogate to minimize the following loss function using stochastic gradient descent:

$$\underset{g}{\arg\min} \ \mathbb{E}_{(\lambda, b, f(\lambda, b)) \sim p_H} \left| f(\lambda_i, b_i) - \left[ g(\lambda_i)_\alpha + g(\lambda_i)_\beta \ b_i^{-g(\lambda_i)_\theta} \right] \right| \quad (5)$$

BO surrogates need to be probabilistic regression models because the acquisition functions require the posterior variance of the predictions. As a result, we train an ensemble of $K$ diverse surrogates $\hat{f}^{(1)}(\lambda, b), \dots, \hat{f}^{(K)}(\lambda, b)$ with the Deep Ensemble strategy (Lakshminarayanan et al., 2017), by initializing each surrogate with different weights and by training with a different sequence of mini-batches. The posterior mean $\mu$ and the posterior variance $\sigma^2$ of the power law ensemble are trivially computed as:

$$\mu_{\hat{f}}(\lambda, b) = \frac{1}{K} \sum_{k=1}^K \hat{f}^{(k)}(\lambda, b),$$

$$\sigma_{\hat{f}}^2(\lambda, b) = \frac{1}{K} \sum_{k=1}^K \left( \hat{f}^{(k)}(\lambda, b) - \mu_{\hat{f}}(\lambda, b) \right)^2 \quad (6)$$

The acquisition function of our approach relies on selecting the configuration $\lambda$ with the lowest estimated loss at the full budget. In other words, we select the hyperparameter configurations that are expected to achieve the best performance at the end of the optimization procedure.

A commonly used acquisition function in the domain is Expected Improvement (EI) which incorporates both the mean and uncertainty of predictions, applying a trade-off between exploration and exploitation. Consequently, in our work, we use the Expected Improvement (EI) acquisition with the estimated full budget's ($b = $ max_budget) posterior mean and variance.

$$\text{EI}(\lambda, b | H) = \mathbb{E} \left[ \max \left\{ \mu_{\hat{f}}(\lambda, b) - f^{(best)}(b = \text{max\_budget}), 0 \right\} \right],$$

$$\lambda^{\text{next}} := \underset{\lambda \in \Lambda}{\arg\max} \ \text{EI}(\lambda, b | H) \quad (7)$$

The best observed loss until a budget $b$ is denoted as $f^{(best)}(b) := \min \{ f(\lambda, b') \mid (\lambda, b', f(\lambda, b)) \in H \wedge b' < b \}$. We briefly define the acquisition in Equation 7, and refer the reader to Mockus et al. (1978) for the details of the EI.

However, after selecting a configuration with our variant of the EI acquisition, we do not naively run it until convergence. Instead, we propose a novel multi-fidelity strategy that advances the selected $\lambda^{\text{next}}$ of Equation 7 by a small budget of $b^{\text{step}}$, e.g. 1 epoch of training. Therefore, the selected $\lambda^{\text{next}}$ will be evaluated at $b^{\text{next}}$ as defined in Equation 8. Notice our proposed strategy also covers new configurations with no learning curve evaluations in $H$.

$$b^{\text{next}} := \begin{cases} b^{\text{step}}, & \nexists \lambda^{\text{next}} : (\lambda^{\text{next}}, \cdot, \cdot) \in H \\ b^{\text{step}} + \underset{(\lambda^{\text{next}}, b, \cdot) \in H}{\max} b, & \text{otherwise} \end{cases} \quad (8)$$

## 5 A Proof-of-concept Example

To visually demonstrate the power law surrogate, we created a 1-dimensional search space where we train a Preact ResNet (He et al., 2016) on the CIFAR-10 dataset (Krizhevsky et al., 2009). We train multiple versions of the model applying a different dropout hyperparameter $\lambda \in [0.1, 0.85]$ for 50 epochs, with a cosine annealed learning rate and an initial value of $10^{-2}$. We train our power law surrogate on a subspace of the hyperparameter search space. We use the full validation curves for the aforementioned hyperparameter subspace, except for the hyperparameter configuration corresponding to 0.65, where, we use only a subset of the learning curve. Ideally, our power

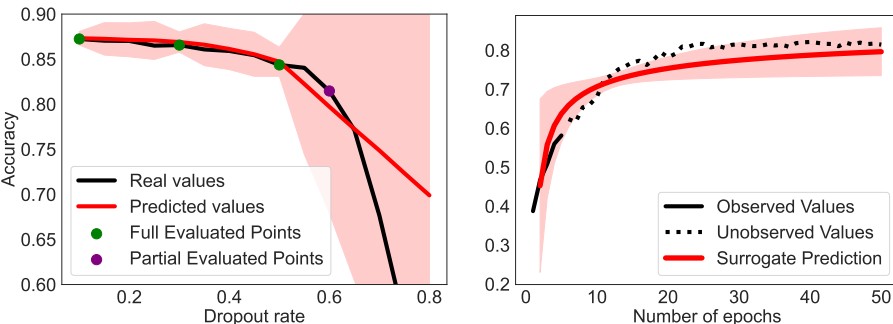

Figure 1: The power law predictions and uncertainty estimations on the CIFAR10 1-D search space. **Left:** The maximal budget across different hyperparameter configurations (dropout rates). **Right:** Across different budgets for the same hyperparameter configuration that was observed partially.

law surrogate should model a low uncertainty in the observed hyperparameter configurations and a higher uncertainty in the partial or unobserved regions.

As shown in Figure 1, our power law surrogate fits the training data well by generalizing correctly across different hyperparameter configurations. Furthermore, the power law surrogate models a low uncertainty in the region of observed hyperparameter configurations, and a higher uncertainty in the region of partially observed configurations that scales more the further we move from the region of observed points. Additionally, Figure 1 (right) conveys that when we evaluate the hyperparameter configuration that is only observed partially, the uncertainty scales with the budget. More concretely, the surrogate is more uncertain the higher the budget (epoch) for which we predict compared to the last observed value.

## 6 EXPERIMENTAL PROTOCOL

In our experiments, we standardize the data by performing min-max scaling for our method and every baseline included. If a baseline has a specific preprocessing protocol, we do not apply min-max scaling but we apply the protocol as suggested by the authors. In the following experiments, we report the regret of the best configuration found $\lambda^{\text{best}}$, which is defined as:

$$R\left(\lambda^{best}\right) = \max\left(0, f^{(oracle)}\left(b = \text{ max\_budget}\right) - f^{(\text{best})}\left(b = \text{max\_budget}\right)\right) \tag{9}$$

where $f^{(oracle)}\left(b = \text{ max\_budget}\right) := \min\left\{f\left(\lambda, b'\right) \mid \left(\lambda, b', f\left(\lambda, b\right)\right) \in H \wedge b' < b\right\}$. In short, the regret is the difference in the metric performance from the best possible hyperparameter configuration (oracle) on the dataset to the best-found hyperparameter configuration by a method . The metric is benchmark-specific, since the benchmarks do not support a common metric. On a dataset level, we report the average regret across 10 repetitions with different seeds. When reporting results over all datasets, we report the averaged normalized regret. The normalized regret for each dataset is calculated by dividing the regret by the difference between the best value and the worst value specific to the dataset. We normalize the regret because over all datasets, there are regret distances of different scales, which in turn can lead to low regret values that can dominate the overall averaged results for all datasets. For more information regarding the detailed implementation of our method, we refer the reader to Appendix A.

### 6.1 BENCHMARKS

**LCBench:** A benchmark that features 2,000 hyperparameter configurations that parametrize the architecture of simple feedforward neural networks, as well as, the training pipeline (Zimmer et al., 2021). The benchmark features 7 numerical hyperparameters. The hyperparameter configurations were evaluated for 51 epochs on 35 different datasets from the AutoML benchmark (Gijsbers et al., 2019). When reporting results for LCBench, we use the balanced accuracy metric.

**PD1:** A deep learning benchmark (Wang et al., 2022) that consists of recent DL architectures run on large vision datasets such as CIFAR-10, CIFAR-100, ImageNet, as well as statistical modeling corpora and protein sequences datasets from the bioinformatics domain. Every search space features varying learning curve lengths, ranging from 5 to 1414, and a different number of evaluated hyperparameter configurations ranging from 807 to 2807. The search space includes hyperparameter configurations that parametrize the learning rate, the learning rate scheduler and the momentum. When reporting results for the PD1 benchmark, we use the accuracy metric.

**TaskSet:** A benchmark that features different optimization tasks evaluated in 5 different search spaces (Metz et al., 2020). For our work, we focus on the Adam8p search space, which is among the largest search spaces in the benchmark with 1000 hyperparameter configurations, every hyperparameter configuration featuring 8 continuous hyperparameters. The hyperparameters control the learning rate, the learning rate schedulers and the optimizer. Every run consists of 50 steps, where, every step corresponds to 200 training iterations. For variety among our benchmarks, we focus on 12 NLP tasks. When reporting results for TaskSet we use the loss metric since the benchmark does not offer information regarding accuracy.

## 6.2 BASELINES

**Random Search:** Randomly samples hyperparameter configurations for the largest possible budget.

**Hyperband:** Uses multiple brackets with different trade-offs of the initial budget and number of epochs to initially train (Li et al., 2017). It then further applies Successive Halving (SH) (Jamieson & Talwalkar, 2016) on every individual bracket to decide which configurations to further train and which configurations to stop training.

**ASHA:** An asynchronous version of SH (Li et al., 2018) that does not wait for all configurations to finish in a bracket before running for the next fidelity.

**BOHB:** An extension of Hyperband that replaces the random sampling of hyperparameter configurations in the initial brackets with model-based sampling (Falkner et al., 2018). BOHB uses TPE (Bergstra et al., 2011) as an inner model and has an individual model for every fidelity.

**DEHB:** An additional extension of Hyperband, which differs from BOHB by using evolutionary strategies to sample the initial hyperparameter configurations (Awad et al., 2021).

**SMAC:** A method that extends Hyperband but uses random forests to sample the initial hyperparameter configurations for a bracket (Lindauer et al., 2022).

**Dragonfly:** We use the Dragonfly Library (Kandasamy et al., 2020) to compare against BOCA (Kandasamy et al., 2017), a multi-fidelity method that uses Gaussian processes to predict the next hyperparameter to evaluate and the fidelity for which it should be evaluated.

For all the baselines, we use their official public implementations. We provide additional details in Appendix C.

## 7 RESEARCH HYPOTHESES AND EXPERIMENTAL RESULTS

**Hypothesis 1:** *The power law assumption improves the quality of the learning curve predictions.*

Initially, we compare power laws against GPs and simple feedforward neural networks by fitting individual instances of every algorithm on a partially observed learning curve for every corresponding hyperparameter configuration. We repeat the procedure with different fractions of partially observed learning curves for every dataset in the LCBench benchmark. We check the accuracy of each model in predicting the final performance by calculating the correlation. Lastly, we investigate the performance of a simple feedforward neural network and Deep Power Laws (DPLs) conditioned on the hyperparameter configurations. In particular, we do not fit a model on the partial learning curve of every hyperparameter configuration, but, we fit a model for all partial learning curves of different hyperparameter configurations, learning a shared model for all configurations.

As can be seen in Figure 2, DPLs manages to have the highest accuracy compared to all the considered model algorithms. Observing the results, fitting an individual power law to a learning curve

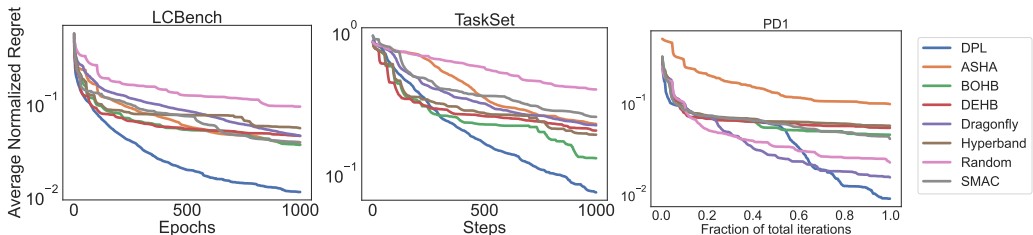

Figure 3: A comparison of all methods considered in the experiments over the number of epochs/steps for all the considered benchmarks. We report the average normalized regret for all methods. A step corresponds to 200 iterations in the case of TaskSet.

achieves predictions of higher quality compared to fitting a simple feedforward neural network or Gaussian processes.

Furthermore, conditioning a neural network to output based on a power law formulation yields a DPL that can share information across hyperparameter configurations while improving accuracy compared to the individual power laws. As expected, the effect is more noticeable in the small data regime, where, we observe only a small subset of observed points from the learning curve. Lastly, a DPL is less expensive to fit on a benchmark compared to fitting an individual power law for every hyperparameter configuration which does not scale and it overcomes the issue of fitting an individual power law, which requires data points for every hyperparameter configuration. Based on the results, we consider Hypothesis 1 to be validated and that **DPLs yield qualitative predictions that surpass different models fitted on individual hyperparameter configurations**.

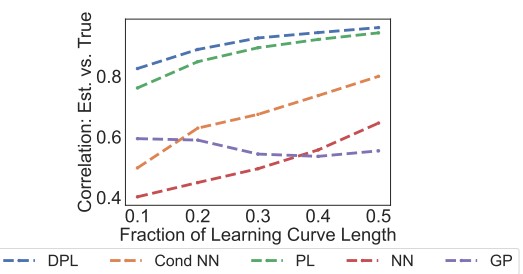

Figure 2: The correlations of individual models, and the DPL, for different fractions of observed learning curves across datasets. **DPL**: Deep Power Law, **Cond NN**: Conditioned neural network, **PL**: Power Law, **NN**: Neural Network, **GP**: Gaussian processes.

**Hypothesis 2:** *Our method DPL achieves state-of-the-art results in HPO.*

Initially, we compare the average normalized regret of our method against all competitors over the number of epochs for all the benchmarks. In Figure 3, we show the performance of all the methods considered in the experiments over the number of epochs, as observed, DPLs manage to outperform in all the considered benchmarks. In the case of LCBench, DPLs converge faster to a better solution compared to the competitor methods and continue to increase the gap in performance until the optimization process ends. Furthermore, we observe the same trend with TaskSet, only in 25% of the optimization process the DPLs converge to a better solution compared to all the considered baselines and increase the lead until the optimization process ends. Lastly, since PD1 features datasets with different learning curves, we consider a budget of 20 full function evaluations for every dataset (roughly the same number of full function evaluations as in the other two benchmarks). We normalize the budget by dividing the number of epochs by the total number of epochs per dataset. As seen in Figure 3, our method converges slower compared to DragonFly, the closest baseline in the benchmark, however, when 75% of the optimization process is reached, our method matches the baseline in performance and continues to improve until the end of the optimization process, once again, performing best compared to all baselines.

In addition to the average normalized regret, Figure 4 provides the critical difference diagrams that show the average dataset ranks of all methods for the different benchmarks, where, DPL consistently achieves the best rank with a statistically significant difference in the majority of cases. In particular, we provide the critical difference diagrams during half the optimization procedure and at the end of the optimization procedure. As observed, for LCBench and TaskSet, DPL achieves the best per-

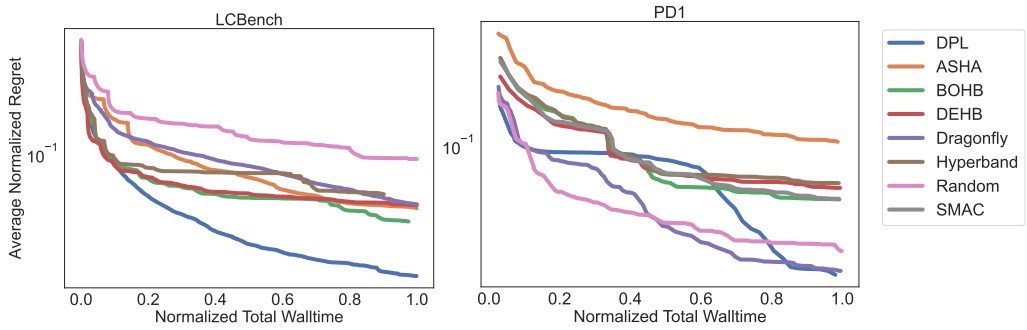

Figure 4: The critical difference diagrams for DPL and the other baselines in all the considered benchmarks. **Top**: The The numbers indicate the average ranks across all datasets (the lower the better). Connected lines correspond to methods whose result difference is not statistically significant.

Figure 5: The performance of rival methods over the normalized time for all the benchmarks considered. We report the average normalized regret for all baselines.

formance with a statistically significant difference in results only halfway through the optimization procedure, retaining the statistical significance until the end of the HPO procedure. In the case of PD1, our method is among the best three methods during half of the optimization procedure, with a non-significant difference compared to the other two top-3 methods. Although at the end of the optimization procedure our method does not have a statistical difference in the results compared to the other methods, it still achieves the best rank over all datasets.

Lastly, we analyse the performance of DPL and all the other methods considered in the experiments in Figure 5, where, as it can be observed, DPL manages to outperform the competitors even when method's overhead time is included, showing that the time overhead of DPL is negligible in the results. In more detail, the total time includes the time to evaluate a hyperparameter configuration and the time taken by each respective method in its inner process to suggest the next hyperparameter configurations to evaluate. Additionally, before the results are averaged for all datasets, the per-dataset time is normalized by the time it took random search to finish the optimization procedure. TaskSet is not included in Figure 5 since it does not offer information regarding the time. Given the results, we conclude that Hypothesis 2 is validated and that **DPL manages to achieve state-of-the-art performance**.

**Hypothesis 3:** *DPL explores the search space more efficiently compared to the baselines.*

We conduct further analyses to understand the source of the efficiency of our method DPL versus the baselines. As a result, we analyze two important aspects, the quality of the evaluated configurations, as well as the exploration capability of our gray-box HPO. Initially, we measure what fraction of the top 1% configurations (ranked by accuracy) our method considers. Figure 6 (left) shows that until convergence our method can discover significantly more top configurations in the datasets of the LCBench benchmark, compared to baselines.

The middle plot of Figure 6 shows the average regret for each configuration promoted to the respective budget. According to this plot, DPL is more efficient by assigning budget only to configurations with lower regret compared to the other methods. The precision and regret plots demonstrate that

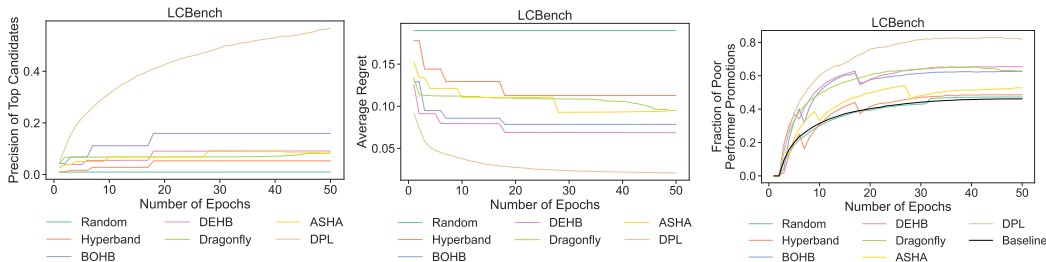

Figure 6: Study of DPL's efficiency over the course of HPO. **Left:** Share of the best candidates selected during training. **Middle:** Average regret of configurations chosen to be trained at each budget. **Right:** Share of top third configurations at a given budget which were bottom two third configurations at a previous budget.

the quality of the evaluated configurations is largely better than all baselines, therefore, giving our method a significant lift in the performance rank. Last but not least, the right plot shows the percentage of configurations that were performing poor in an earlier epoch (i.e. accuracy-wise in the bottom $2/3$ of configurations up to the epoch indicated at the x-axis) but performed better at later epochs (i.e. at the top $1/3$ of configurations). Furthermore, we added a line labeled with "Baseline", which represents the fraction of previously poor-performing configurations of all configurations. Such behavior is observed often with learning curves, for instance, strongly regularized networks converge slowly. The results indicate that our method can better explore unpromising early configurations, by giving them a chance through the uncertainty estimation of our ensemble, and the respective Bayesian optimization mechanism.

## 8 LIMITATIONS

Our proposed ensemble of power law functions achieves an important gain in performance concerning the state-of-the-art in gray-box HPO and highlights the efficiency of modeling learning curves with the power law assumption. However, we believe that further research is needed to calibrate the power law model for the beginning and the end parts of the learning curves. Recent work highlighted that the error rate at very small fidelities (e.g. after a few mini-batches) is not a power law (Rosenfeld et al., 2021). Contrary to the common expectations, we experienced that the uncertainty estimation arising from the Deep Ensemble (Lakshminarayanan et al., 2017) is not very qualitative compared to standard BO surrogates such as Gaussian Processes. In addition, having to train an ensemble has additional computational costs, due to the necessity of training multiple power law models. In the future, we plan to re-conceptualize our surrogate, by combining power laws with Gaussian Processes.

## 9 CONCLUSIONS

In this work, we introduce Deep Power Law (DPL), a probabilistic surrogate based on an ensemble of power law functions. The proposed surrogate is used within a novel gray-box HPO method based on Bayesian optimization. In contrast to the prior work we exploit the recently-discovered scaling laws for estimating the performance of Deep Learning models. Through extensive experiments comprising 7 baselines, 57 datasets, and search spaces of diverse deep learning architectures, we showed that DPL outperforms strong HPO baselines for DL by a large margin. As an overarching contribution, we advanced the state-of-the-art in the important field of HPO for Deep Learning.

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

## A  IMPLEMENTATION DETAILS

For our method, we use a 2-layer feedforward neural network with 128 units per layer. Additionally, we use batch normalization after every linear layer. We use Leaky ReLU as our non-linearity. Our network has 3 output units, which, are then combined to yield the power law output. We apply the Sigmoid non-linearity activation only on the $\beta$ and $\gamma$ output units. For the experiments in the pd1 benchmark, we apply a Leaky ReLU non-linearity to improve convergence, since, the benchmark features low error rate values. We use PyTorch version 1.12 as the main library on top of which we build our method.

We use the MSE loss to train our network, coupled with Adam featuring an initial learning rate of $10^{-3}$. We train every network of our ensemble for 250 epochs only in the beginning of the hpo optimization process and we continuously refine the model for 20 epochs every hpo iteration. Lastly, we use 5 models to build our ensemble of DPLs. For the experiments, we use an initial history H of 1 randomly sampled hyperparameter configuration evaluated for 1 epoch.

## B  BENCHMARKS

**LCBench**    We use the official implementation as the interface for the LCBench benchmark [1]. As suggested by the authors, we use the benchmark information starting from the second step and we skip the last step of the curve since it is a repeat of the preceding step.

**TaskSet:**    The TaskSet benchmark features 1000 diverse tasks. We decide to focus on only 12 NLP tasks from the TaskSet benchmark to add variety to our entire collection of datasets. Our limitation on the number of tasks included is related to the limited compute power, as we are unable to run for the entire suite of tasks offered in TaskSet. TaskSet features a set of 8 hyperparameters, that consists of i) optimizer-specific hyperparameters, such as the learning rate, the exponential decay rate, $\beta_1$ and $\beta_2$, and Adam's constant for numerical stability $\varepsilon$, ii) hyperparameters that control the linear and exponential decay schedulers for the learning rate decay, and lastly iii) hyperparameters that control the L1 and L2 regularization terms. Every hyperparameter in TaskSet except $\beta_1$ and $\beta_2$ is sampled logarithmically.

**PD1:**    We use the synetune library (Salinas et al., 2022) for our interface to the PD1 benchmark. From the benchmark, we only include datasets that have a learning curve of length greater than 10. We furthermore only include datasets that have a learning curve lower or equal to 50 to have a fair comparison between all benchmarks by having approximately 20 full function evaluations. PD1 features 4 numerical hyperparameters, $lr\_initial\_value$, $lr\_power$, $lr\_decay\_steps\_factor$ and $one\_minus\_momentum$, where $lr\_initial\_value$ and $one\_minus\_momentum$ are log sampled. The learning rate decay is applied based on a polynomial schedule.

## C  BASELINES

**Random Search:** We implemented random search by randomly sampling hyperparameter configurations from the benchmarks with the maximal budget.

**Hyperband, BOHB, LCNet:** We use version 0.7.4 of the HpBandSter library as a common codebase for all 3 baselines [2]. For the last approach mentioned, despite heavy hyperparameter tuning of the method, we could not get stable results across all the benchmarks and hence dropped the method from our comparison.

**ASHA:** For the implementation of ASHA we use the public implementation from the optuna library , version 2.10.0.

**DEHB:** We use the public implementation offered by the authors [3].

**MF-DNN:** In our experiments we used the official implementation from the authors [4]. However, the method crashes which does not allow for full results on all benchmarks.

**SMAC:** For our experiment with SMAC we used the official code base from the authors [5].

**Dragonfly:** We use version 0.1.6 of the publicly available Dragonfly library.

For all the multi-fidelity methods considered in the experiments, we use the same minimal and maximal fidelities. In more detail, for the LCBench and TaskSet benchmarks we use a minimal fidelity lower bound of 1 and a maximal fidelity lower bound equal to the max budget. In the case of PD1, where, the learning curves have different lengths, we use a minimal bound that allows for a maximal amount of 4 brackets, if no more than 4 brackets can be achieved we use a minimal budget of 1.

## D  PLOTS

---

[1]https://github.com/automl/LCBench
[2]https://github.com/automl/HpBandSter
[3]https://github.com/automl/DEHB/
[4]https://github.com/shib0li/DNN-MFBO
[5]https://github.com/automl/SMAC3

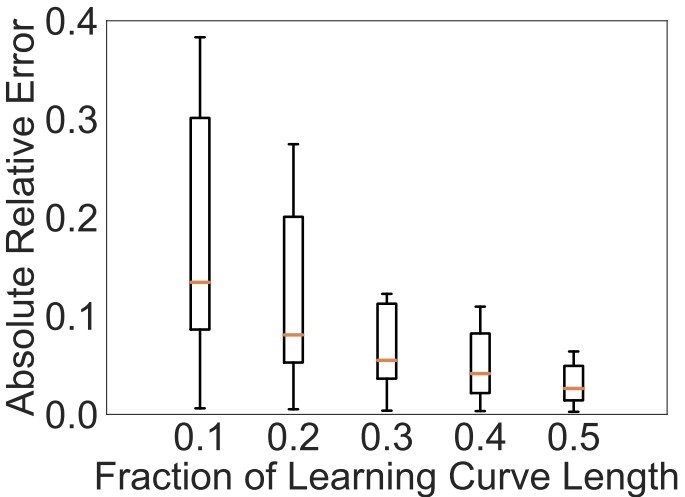

Figure 7: The dataset absolute relative error distribution of DPL over the different learning curve fractions. The distribution is calculated from the ground truth and prediction values, averaged over all configurations of a dataset.

