# OpenReview forum: "Deep Power Laws for Hyperparameter Optimization"
_ICLR.cc/2023/Conference — Submitted to ICLR 2023_

### Official Review · Reviewer_vKZP · 2022-10-17

**Confidence:** 4
**Correctness:** 2
**Technical Novelty And Significance:** 2
**Empirical Novelty And Significance:** 3
**Recommendation:** 5

**Clarity, Quality, Novelty And Reproducibility:**

The paper is clear, although leaves more than desired for the reader to devine in terms of the experimental results (this reader found himself referring to the code for clarifications on the implementation of the EI, which is not ideal — as the paper should be self contained).

For quality, as mentioned above, the experimental design and analysis need to be improved.

If improved such that a robust and practical significant gain in hyperparameter optimization may be demonstrated, this would be a sufficiently significant novel contribution.

Reproducibility given the code is adequately addressed by the code, and I commend the authors for providing it — while some more documentation (starting with a non-empty readme) and cleanliness would be warmly advised to improve clarity, even orthogonal to this review…

Code bits like hamper clarity:
(excerpt from surrogate_models/power_law_surrogate.py:  )
{
if self.iterations_counter == 200:
            b = 6
}


Finally, the paper would do well to engage more deeply with prior work.

Pertinent to the assumption of powerlaw training curves the work of Pretrum et. al [1] comes to mind.

Further, optimization hyper-parameter scaling with model and data scale (a very important canonic tool in practice), should be mentioned — i.e. batch size scaling, Kaplan et. al. [2]

[1] https://arxiv.org/pdf/2010.08127.pdf

[2] https://arxiv.org/pdf/2001.08361.pdf


**Strength And Weaknesses:**

$\bf{Strengths: }$

Scaling laws and specifically their power law characteristics have emerged as a powerful tool for both practical and theoretical investigation. Attempting to apply this very strong prior for improving in a principled baysian framework for the gains in the efficiency of hyperparameter search is an important and potentially very useful direction.

The preliminary results in the paper are encouraging relative to current methods examined in some of the cases presented in terms of performance (though not convincingly in all — e.g. no statistically significant difference between the random baseline and DPL as shown in figures 4).
Similarly, the claimed efficiency (and demonstrated relative to waltime — figure 5) is of paramount importance, as at the large scale hyperparameter tuning becomes practically prohibitive.

$\bf{Weaknesses: }$

However, there are several methodological improvements which merit attention:

1. The core assumption of power-law learning curves, is poorly qualified.

It is still very often the case, in practice, that multiple schedules are involved in the training configuration. These, in particular, are subject of (hyperparameter) tuning and materially violate the powerlaw assumption. To name several such very common schedules one may consider warmup and learning rate scheduling (stepped or otherwise as in this paper where cosine scheduling with no repetition is applied at least in part).

The proof of concept provided on CIFAR10 suffers from several limitations, not contributing to the further the confidence in the power-law assumption:
The uncertainty of the estimation diverges very quickly outside the interpolation area (area within the range of measurements) — figure 1 left. Further, the learning curve itself *does not* agree with the prediction, even on the measured area, further indicating that it is probably not actually a powerlaw in this case (as, by construction, the prediction is a powerlaw).

2. The analysis conducted in figure 2, does show the relative superiority of the correlation, however correlation may be too blunt an instrument for the purposes of interest. High correlation may be present with significant (perhaps even rank non-conservation) deviations at the accuracies of interest (where small absolute error differences are large relative differences, in the low error regime).

These limitations are not merely of aesthetics, qualifying when the results of this method may be trusted is very important in order to substantiate it as a good hyperparameter optimization method.

For example — from what point in the learning curve is the approximation reliable? (and consequently, what is the expected computational/budget gain?) . and what limitations of applicability does this method impose with respect to the optimization protocol?

At the least a relative deviation (e.g. error in log scale correlation) should be added, and with a measure not only of the average over all datasets, but rather a breakdown per dataset and with both average (over learning curve agreement with different hyperparameters) and deviations.
As well as a discussion and examples of where this prior actually does not agree with the learning curves and fails.

3. Evaluation of performance:

The authors use existing benchmarks, but show average regret or average normalized regret over a multitude of datasets. It is not clear from such an analysis what is the individual dataset performance or whether the standard deviation is very large as to hide phenomena which may adversely affect the performance of the proposed method in different scenarios. For example, a behavior which makes one wonder as to its origin is figure 3, PD1 average normalized regret which is worse than the random baseline up until >0.5 of iterations.

This same concern is mirrored in the rank statistical parity of the random baseline even at full budget (figure 4, right).

Can it be explained in these cases why the method does so poorly (even if others are as poor).

Again, this is of practical importance --- hyper parameter optimization is an exercise which one engages in the context of a specific, single task. A method should be shown superior separately on all, or called out for its failure modes when not universally applicable.

4. Finally, it is unclear to this reviewer if this method has strong predictive power in the sense that it can be effectively used with a low budget in order to reliably precede large scale (prohibitively expensive) experiments. The benchmarks used may answer this question, but it remains opaque in the body of the work.
Specifically, since the DPL requires training on multiple degrees of freedom (hyperparameter dimensions) in order to form a reliable prediction along these degrees of freedom, to what extent is this done at small fidelity? What is the expected computational gain relative to current methods which involve e.g. grid search at the small scale and then hyperparameter scaling?

It is crucial (especially if this work remains in the purely empirical domain, as is currently governed by the NN at the heart of the hyperparameters-loss estimation), that the computational gain be fleshed out for this venue to offer a path towards superior hyperparameter optimization.


**Summary Of The Paper:**

The authors address the very important topic of hyperparameter selection, by marrying scaling laws and the Bayesian optimization hyperparameter search --- under the assumption of learning curves taking a powelaw form.

The central assumption in the paper is that training curves, and in particular validation error as function of number of epochs, are well approximated by powerlaws, with the coefficients dependent on the hyperparameter selection.

Based on this assumption the authors propose a bayesian optimization method for hyperparameter search which uses a neural network (NN) estimator of the expected coefficients value of unseen hyperparameters.
The NN is trained on a batch of hyperparameters pre-trained configurations (learning curves), multiple times, with seed and sample order permutations, to form an ensemble from which mean and variance are predicted.

The authors showcase a preliminary demonstration on a CIFAR10 task of predicting the performance while scanning dropout level and show consistent uncertainty behavior with respect to the distance (in dropout level, and in number of epochs) between measured and predicted points.

The authors then continue to assess the validity of the framework and its performance on 3 benchmarks, relative to several alternative hyperparameter optimization baselines.

First, the agreement of the powerlaw-base learning curve is assessed relative to other learning curve prediction methods. Superior relative correlation is demonstrated.

Second, the authors examine the averaged regret of the proposed method relative to the baselines. And finally the authors examine the exploration / exploitation of the method.


**Summary Of The Review:**

Hyperparameter efficient optimization remains one of the practical/economical thorns in the field. The paper offers an interesting direction of capitalizing on predictable phenomena in the form of scaling laws for the leveraging in the context of bayesian inference for hyperparameter search.

While preliminary results are encouraging, there are numerous weaknesses pertaining to the robustness of the underlying assumptions, performance and practical (namely compute) expected utility.

Significant work is required for addressing these above mentioned issues, and I will be happy to upgrade my review and score if they are adequately resolved.

%%%%%% updated score following rebuttal clarifications %%%%%%
see reply for details --- core issue is concern with respect to actual applicability due to mixed results. Strongly encourage to resubmit with demonstration of the utilization of method claimed superiority to improve performance in the wild

---

> ### Author Response · Authors · 2022-11-18
> **Response to Reviewer vKZP Part 1**
>
> We thank the reviewer for investing time in providing a lengthy and extensive feedback and for additionally finding our work novel and of high importance. Below we address your concerns:
>
> **With regards to missing statistical significance in Figure 4:**
>
> To not become repetitive, we would like to point out to the reviewer that a non-connecting line indicates that the method performance is statistically significant compared to the other methods. With regards to LCBench and TaskSet, we are better statistically significantly compared to the other methods at the end of the optimization process, but we are also statistically significantly better at half of the optimization procedure, which **equals circa 10 full function evaluations**. Only in the case of PD1 we are not statistically significant compared to the other methods, however, **we are still the method with the best per-dataset rank.**
>
> **With regards to the core assumption of the power laws being poorly qualified:**
>
> We would kindly invite the reviewer to read the response to Reviewer KFvH regarding the learning curve schedules.
> Regarding the uncertainty modeling, in the Limitations section 9 we have already stated that the modeling of uncertainty from the deep ensemble has its drawback, especially, in a multi-fidelity scenario and we have suggested a potential improvement by using Gaussian Processes.
>
> We agree with the reviewer that there are regions where the power law might not fit the curve perfectly (as stated in Section 9), however, it is a very good approximation for it as indicated by our extensive experiments, further tweaking the assumption to handle the parts where the power law assumption breaks, could only improve the results of our method.
>
> Furthermore, we would like to point out, that our power law formulation: $ \hat f\left(\lambda, {b}\right) := g(\lambda)_\alpha + g(\lambda)_\beta * {b}^{-g(\lambda)_\gamma} ,   g: \Lambda \rightarrow R^3$
>
> is able to handle cases where the configuration's curve does not improve by pushing $g(\lambda)_\beta$ to zero and by modeling the best fitting curve/line with $g(\lambda)_\alpha$. We have observed the aforementioned behavior in LCBench.
>
> **With regards to the correlation:**
>
> We believe that correlation is among the best instruments to indicate how the method keeps the ranks over all the configurations. For example, one might consider a scenario where the deviation from the prediction of the configurations might be low, however, the ranks might not be preserved. If the rank is preserved, then the specific value of a hyperparameter configuration would not matter since if it is ranked higher than the others, it will still be chosen. The only constraint would be not to underestimate consistently for all hyperparameter configurations since it would hinder the acquisition function and as such, the whole optimization process.
>
> We additionally provide the distribution of the dataset absolute relative errors over the different learning curve fractions as requested by the reviewer in Figure 7 in Appendix D.
>
> **With regards to the evaluation metric and the dataset-specific performance:**
>
> We kindly point the reviewer to  Figure 4, which does not show only the statistical significance but also the ranks of the methods on every dataset, by providing the average ranks over the datasets for every method. As can be seen, our method achieves the best rank compared to all the considered baselines.
>
> We will additionally provide the plots for all datasets and methods along with their standard deviations.
>
> **Regarding the fidelity-performance correlation:**
>
> In Figure 2, as observed by the reviewer we do provide information regarding the correlation levels that DPL (and other algorithms) achieve when trained with 10-50% of the learning curve. Additionally, in Figure 7 in Appendix D, we provide more information regarding the absolute relative error that DPL yields based on the learning curve fraction used.
>
> We would like to point out that in our experiments we start with only 1 random hyperparameter configuration evaluated for 1 epoch and the DPL is not pretrained, every other hyperparameter configuration is sampled from the surrogate model.
>
> Based on the above, following Figure 3, in LCBench, we can see that after 250 epochs, which are equal to 5 full function evaluations, our method is matching the best competitor in performance, approximately, 4 times faster. Additionally, only about 500 epochs in, we are matching the final performance of the best competitor, 2 times faster in the case of TaskSet. Lastly, only about 700 epochs in, we are matching the final performance of the best competitor, being 1.43 times faster for PD1. As our method starts with only one hyperparameter configuration and then it iteratively trains, it will only get better in the prediction quality as more configurations are available, the choice of the number of iterations is up to the practitioner based on the final performance requested.

---

> > ### Author Response · Authors · 2022-11-18
> > **Response to Reviewer vKZP Part 2**
> >
> > **Regarding the speed/walltime:**
> > As requested by the reviewer, we provide the following statistics regarding our training time:
> > Our surrogate model (whole ensemble) requires the following time to fit a batch of examples:
> > |Benchmark   | Time (seconds)  | Batch Size  |
> > |---|---|---|
> > |  LCBench |  0.034 | 32  |
> > |  TaskSet | 0.034  | 32  |
> > |  PD1 | 8.08e-06  | 32  |
> >
> > As can be observed, the runtime overhead of our method is negligible. We would like to point out that all the provided results are on the CPU, so a considerable speed-up will be achieved when run on a GPU. We would like to reinforce that we refine our models (20 epochs to be exact) and we do not train from scratch at every BO iteration.
> >
> > Furthermore, for the wall-time plot provided in Figure 5, all the method times are normalized per dataset with the time it took random search to finish the optimization process. The end result is capped over the time it took random search to complete.
> >
> > We thank the reviewer for praising us on open-sourcing the implementation and for spotting an artifact. We will have an updated and well-documented code by the end of the discussion stage with a README that describes in detail how to run the experiments.
> >
> > We believe to have adequately resolved all the uncertainties in the work raised by the reviewer and we would kindly ask the reviewer to update his score based on the feedback. In case the reviewer has more suggestions or doubts, we are happy to answer them.

---

> > ### Comment · Reviewer_vKZP · 2022-11-23
> > **Reply to authors**
> >
> > I thank the authors for the explanations.
> > While part of the issues raised above were made clearer by the authors, as it stands I have lingering topics which have not been adequately addressed. I thus raise my score marginally, however, the paper does not yet meet the acceptance criteria.
> >
> > In particular, the non-powerlaw behavior (as stated as observed by the authors in LBench), coupled with the LBench vs PD1 performance discrepancies (i.e. PD1 figure 5, PD1 above mentioned statistical significance parity with the *random* baseline which was not adressed by the authors response) leave open the question as to the actual reliability/generelizability of the method.
> >
> > Figure 6 highlights only LBench and is notably absent of PD1.
> >
> > I would encourage the authors to re-submit the paper with some cleaning up as to its self contained readability and more convincing utility --- as it shows potential promise.
> >
> > Specifically, it would making the paper much more compelling if the claimed HPO superiority would be put to use in an example showing improvements in the wild outside of the above benchmarks.
> >
> > With the claimed efficiency gains relative to other HPO methods, and the prohibitive optimization space and costs governing contemporary models governing SOTA (e.g. LLMs), this method, if its promise holds, may provide for demonstrating potentially better hyper parameter choices. It would suffice to show that these gains manifest in superior scaling.

---

> > > ### Author Response · Authors · 2022-12-01
> > > **More detailed response to Reviewer vKZP**
> > >
> > > We thank the reviewer for his valuable feedback in helping us improve our paper and we provide further results to convince the reviewer regarding the superiority of our method and the overall experiments.
> > >
> > > **Missing statistical significance in PD1:**
> > >
> > > We have discovered an issue in the conversion script for the pd1 results.  We have rerun our experiment by rerunning all methods and we share the following results with the reviewer:
> > >
> > > The regret over the number of epochs:
> > >
> > > https://anonymous.4open.science/r/DeepRegret-0F61/plots/updated_comparison_epochs_pd1.pdf
> > >
> > > The regret over time:
> > >
> > > https://github.com/ArlindKadra/DeepRegret/blob/development/plots/updated_comparison_time_pd1.pdf
> > >
> > > The significance at 50% of the optimization and 100% of the optimization:
> > >
> > > https://anonymous.4open.science/r/DeepRegret-0F61/plots/updated_pd1_diagram_half.pdf
> > >
> > > https://anonymous.4open.science/r/DeepRegret-0F61/plots/updated_pd1_diagram_full.pdf
> > >
> > > As can be seen, now our method performs better compared to all other methods and has a better convergence, further improving what we provided before by yielding good any-time results. We would like to emphasize that **we have rigorously checked and debugged our benchmarks to investigate the conversions.**
> > >
> > > We would like to additionally point out that **not having statistical significance against random search, is not a weakness of our method, as it is shared by all baseline methods**. This is a scenario that **has been observed in the community, especially in the NAS domain**, where random search is known to perform strongly. Additionally, **statistical significance is a hard constraint to satisfy and the lack of it in this search space is shared by all strong competitors**.
> > >
> > > We believe the results are of interest to the reviewer and should furthermore assert the superiority of our method for the reviewer, as the reviewer requests.
> > >
> > > **Efficiency plots presented only for LCBench, PD1 and TaskSet are missing:**
> > >
> > > We only included LCBench in Figure 6 because of space constraints. We provide the following plots regarding TaskSet and PD1:
> > >
> > > **TaskSet**
> > >
> > > https://anonymous.4open.science/r/DeepRegret-0F61/plots/updated_avg_regret_taskset.pdf
> > >
> > > https://anonymous.4open.science/r/DeepRegret-0F61/plots/updated_precision_taskset.pdf
> > >
> > > https://anonymous.4open.science/r/DeepRegret-0F61/plots/updated_promotions_taskset.pdf
> > >
> > > **PD1**
> > >
> > > https://anonymous.4open.science/r/DeepRegret-0F61/plots/avg_regret_pd1.pdf
> > >
> > > https://anonymous.4open.science/r/DeepRegret-0F61/plots/precision_pd1.pdf
> > >
> > > https://anonymous.4open.science/r/DeepRegret-0F61/plots/promotions_pd1.pdf
> > >
> > > The reviewer can observe the same pattern as in LCBench, where our method explores the search space more efficiently.
> > >
> > > **Prohibitive optimization space:**
> > >
> > > We would like to point out to the reviewer that although the search spaces included in our experiments in Section 7 are finite, the search spaces are not exhaustive and it’s hyperparameter configurations span a very large hyperparameter space,  as such, they are not prohibitive.
> > >
> > >
> > > **The non-power law behavior:**
> > >
> > > We would like to remind the reviewer that for LCBench **in our previous reply, we described a case where there was a non-power law behavior and our formulation of the power law could approximate correctly**, as also observed jointly by the provided rank correlations in Figure 2.
> > >
> > > As a reminder, the case involved a diverging learning curve, which given our formulation:
> > >
> > > $ \hat f\left(\lambda, {b}\right) := g(\lambda)_\alpha + g(\lambda)_\beta * {b}^{-g(\lambda)_\gamma} ,   g: \Lambda \rightarrow R^3$
> > >
> > > Is modeled correctly by pushing $g(\lambda)_\beta$ to zero and by modeling the curve/line with $g(\lambda)_\alpha$. **We did not state that our formulation had trouble representing it (what the reviewer quotes as a concern),**
> > >
> > >
> > > We believe in having clarified all the concerns raised by the reviewer and we would like to invite the reviewer to update the score based on the given clarifications. If there are more questions we would be happy to answer them.

---

### Official Review · Reviewer_KEEA · 2022-10-26

**Confidence:** 1
**Correctness:** 3
**Technical Novelty And Significance:** 3
**Empirical Novelty And Significance:** 3
**Recommendation:** 6

**Clarity, Quality, Novelty And Reproducibility:**

The motivation is clear and the proposed method using scaling law property looks novel. They provided the code and implementational details.

**Strength And Weaknesses:**

- Strengths
  - The writing is clear and easy to follow.
  - Applying scaling law property of learning curves for HPO is novel.
  - This work demonstrated their method on extensive experiments under various domain such as image, tabular, and NLP domains.

- Weaknesses
  - In the experimental section, I think that the title 'Hypothesis' is not appropriate. For example, even if this work demonstrated the performance of their work empirically not theoretically, they regarded Hypothesis 'Our method DPL achieves state-of-the-art results in HPO' is achieved.
  - With the most experiments, this work focus on 'regret' metric. Could the authors show the experiments with the metric 'accuracy'? For example, by applying the proposed method and multiple HPO methods on ResNet and cifar10/imagenet, we can see the model performance.

**Summary Of The Paper:**

This work propose a Deep Power Law (DPL) which uses the scaling law property of learning curves for hyperparameter optimization, by improving fidelity of HOP for deep learning. DPL ensembles predicts the performance of hyperparameter configurations in low-budget regimes as a probabilistic surrogator for Bayesian optimization (BO). This work demonstrated the effectiveness of the proposed method by conducting extensive experiments on 62 datasets under various domain such as image, tabular, and NLP domains.

**Summary Of The Review:**

They uses scaling law property for improving the fidelity of HPO and demonstrated the proposed method on multiple HPO benchmarks.

---

> ### Author Response · Authors · 2022-11-18
> **Response to Reviewer KEEA**
>
> Thank you for the thoughtful review and for finding our work novel and our experiments extensive. Below we address your concerns:
>
> **In regards to the hypothesis naming:**
>
> We understand the reviewer’s confusion, however, we believe hypothesis to be the correct term. For example, when evaluating for statistical significance, we state that we reject or accept the null hypothesis. In the same regard, we accept or reject the hypotheses that we state based on the experimental findings.
>
> **With regards to the regret metric and accuracy:**
>
> The experiment that the reviewer suggests is already part of our work (Hypothesis 2, Section 7). In more detail, the plots regarding LCBench and PD1 indicate the accuracy performance.  To not become repetitive, we invite the reviewer to read the response to Reviewer 3aYJ regarding the average normalized regret point and his suggested experiment. We have further refined Sections 6 and 6.1 to further improve the clarity of our work.
>
> We believe to have clarified all the concerns from the reviewer and we would kindly ask the reviewer to update his score based on the provided feedback.

---

### Official Review · Reviewer_KFvH · 2022-10-27

**Confidence:** 4
**Correctness:** 3
**Technical Novelty And Significance:** 2
**Empirical Novelty And Significance:** 2
**Recommendation:** 3

**Clarity, Quality, Novelty And Reproducibility:**

## Clarity
The paper is clearly written for the most part but some of the details I was looking for was in the appendix or in referenced papers (e.g. description of the surrogate models and search spaces studied in each of the benchmarks).  I suggest the authors include the missing search space specifications in the appendix so the paper is more self contained.
## Quality
The quality is sufficient overall.  I found the multi-fidelity aspect of the DPL algorithm very simple and straightforward, which is okay but this aspect of the algorithm was oversold in the abstract and introduction.
## Reproducibility
I believe sufficient information is provided to reproduce the results for DPL shown in the experiments.

**Strength And Weaknesses:**

Pros:
- The empirical results are strong on the 3 benchmarks studied.
- Paper is well written and easy to understand.

Cons:
- Empirical analysis limited to pretrained benchmarks with a finite number of trained configurations.  In truly continuous search spaces, it is unclear how often DPL will repeat the same hyperparameter setting to be trained for more batches without additional discretization.
- From what I could tell, experiments do not contain search spaces with step wise learning rate schedules where learning rates decrease drastically at fixed intervals.  These types of LR schedules often result in learning curves that do now follow the power law and are used frequently in practice.
- DPL is limited in novelty since power law based performance prediction methods have been studied before and the multi-fidelity aspect just allocates a small unit of work per decision step.

**Summary Of The Paper:**

This paper proposes a multi-fidelity Bayesian optimization approach to hyperparameter optimization called Deep Power Law (DPL) that exploits the power law assumption of learning curves to quickly identify strong hyperparameter settings.  Results on 3 benchmarks with tabular, image, and NLP datasets show DPL to outperform 7 state-of-the-art competitors.

**Summary Of The Review:**

While DPL outperforms many leading HPO methods on the 3 studied benchmarks, the experiments do not sufficiently establish DPL as a viable general HPO method due to missing experiments on non-lookup based benchmarks and lack of search spaces with step LR schedules.  Additionally, the novelty of DPL is limited since it is similar to prior power law performance prediction methods and has a very simple fixed allocation multi-fidelity component.

---

> ### Author Response · Authors · 2022-11-18
> **Response to Reviewer KFvH**
>
> Thank you for your thoughtful review, for finding our paper well-written and for finding our empirical results strong. Below we address your concerns:
>
> **In regards to a continuous search space:**
>
> We understand the reviewer's concern regarding the application of our method in a continuous search space, however, we believe that the potential problematic scenario the reviewer is suggesting is an issue with all multi-fidelity methods in general and it is not specific to our proposed method. It is not certain how often the methods might suggest the same configuration between the low and high fidelity levels.
>
> **In regards to step-wise learning rate schedules:**
>
> We agree with the reviewer that step-wise learning rate schedules are used in the community. In our proof of concept example in Section 5 with CIFAR10, we do use a step-wise cosine annealed learning rate scheduler and as can be seen from the mean prediction and uncertainty, our method is able to generalize well.
>
> Additionally, we would like to point out that TaskSet does feature a linear and exponential learning rate scheduler and our method outperforms all competitors statistically significantly not only at the end of the optimization process but also in the middle of it, corresponding to circa 10 full function evaluations. Furthermore, PD1 does feature a polynomial step-wise learning rate scheduler with different factors.
>
> Following the feedback from the reviewer, we have updated Section 6.1 and Appendix B with more detailed information about the hyperparameters of individual benchmarks.
>
> **In regards to novelty:**
>
> To the best of our knowledge, this is the first time where the power law coefficients are conditioned on a parametric neural network and are applied successfully via a Deep Ensemble on a Gray-box HPO setting yielding state-of-the-art results.
>
> Based on the clarifications given, we believe to have addressed all the reviewer's concerns and we would kindly ask the reviewer to update his score based on the feedback given. In case there are more questions, we are happy to answer them.

---

> > ### Comment · Reviewer_KFvH · 2022-11-22
> > **Post Author Response**
> >
> > I have reviewed the author response and will maintain my score.  The authors did not adequately address the weaknesses I brought up around handling of continuous hps and step wise learning rate decay. More explicitly, I would expect to see results on a few search spaces with continuous hyperparameters that are not tabular lookup benchmarks and also search spaces with step lr that decreases by say a factor of 10 after progressing 80% of the way through training.  Given the prevalence of transformer architectures, I would also like to see some results for that family of models if not already included.  In the absence of such experiments, I do not believe the paper is in a state for acceptance.

---

> > > ### Author Response · Authors · 2022-11-30
> > > **More detailed response**
> > >
> > > We thank the reviewer for his response regarding our initial reply and we provide a more detailed clarification:
> > >
> > > **The continuous search space:**
> > >
> > > The reviewer mentions the number of times that the same hyperparameter configuration is recommended as a potential drawback of our method in a continuous search space. As stated previuously, we believe this might be a potential problematic scenario, however, it is a **potential problematic scenario for all multi-fidelity methods and not unique to our method**. For example:
> > >
> > > Considering the recent multi-fidelity methods that are widely used in the community that extend hyperband, we denote a hyperband bracket by $s$ (where successive halving (SH) is applied inside bracket $s$). Then, the initial population with hyperparameter candidates $\Lambda_s$ of the bracket $s$ is generated by sampling from a model in a continuous search space. There is no connection from bracket $s$ to bracket $s+1$ that use different fidelity levels and based on that, the model based sampling can suffer from the same issue that the reviewer mentions by considering hyperparameter configurations that have not been sampled previously, for which we basically have no learning curve.
> > >
> > > As such, we ask the reviewer, is this potential problematic scenario not common for all multi-fidelity methods? If the reviewer disagrees, we would kindly ask the reviewer for his\her explanation.
> > >
> > > **The suggested experiment with the learning rate decay:**
> > >
> > > We would like to point out to the experiment results in Section 7, which already contain the scenario that the reviewer asks. In more detail, the pd1 search space,  features a polynomial learning rate scheduler (Equation 6 in [1]). The scheduler is parametrized based on the decay step fractions $\lambda$, the step $\tau$, the total optimization steps $T$, the power $p$, the initial learning rate $\eta$ and the 1 - momentum 1 - $B$ as following:
> > >
> > > $\eta_t = \frac{\eta}{1000} + (\eta - \frac{\eta}{1000}) (1- \frac{min(\tau, \lambda T)}{\lambda T})^p $
> > >
> > >
> > > As an example, by considering a learning rate value $\eta$ of 0.1, a $ \lambda$ value of 0.8, and a $p$ of 0.1, we would get the following learning rate trajectory as shown in the plot:
> > >
> > > https://anonymous.4open.science/r/DeepRegret-0F61/plots/pd1_schedule.pdf
> > >
> > > Which is exactly what the reviewer asks, a learning rate that decays drastically around 80% of the optimization.
> > >
> > > **Considering transformer architectures in the results:**
> > >
> > > We would like to point out to the the reviewer that the results shown in Section 7 already cover transformer architectures, we point the reviewer to Table 2 of [1] for a more detailed description of all the considered models in our search space.
> > >
> > >
> > > [1] Wang, Zi and Dahl, George E. and Swersky, Kevin and Lee, Chansoo and Mariet, Zelda and Nado, Zachary and Gilmer, Justin and Snoek, Jasper and Ghahramani, Zoubin
> > > Pre-trained Gaussian processes for Bayesian optimization
> > >
> > > We believe to have clarified in more detail the concerns raised by the reviewer and we would kindly invite the reviewer to upgrade the score based on the given clarifications. If there are more questions, we would be happy to answer them.

---

### Official Review · Reviewer_3aYj · 2022-10-31

**Confidence:** 3
**Correctness:** 2
**Technical Novelty And Significance:** 3
**Empirical Novelty And Significance:** 2
**Recommendation:** 5

**Clarity, Quality, Novelty And Reproducibility:**

The clarity and quality of the paper is not good, as explained in the weakness section.

The proposed method sounds pretty novel though, as it is the first to leverage the power law in learning curves.

Since quite a few important parts are missing, such as deep ensemble and expected improvement, it is hard to reproduce the results based on the description in the paper.

**Strength And Weaknesses:**

Strength:

1. Adopting the observation that the performance of machine learning methods as a function of budgets usually follows a power law to hyperparameter optimization is quite novel.
2. The performance of DPL seems to be better than most of the competitors in terms of average normalized regret in several benchmarks, e.g. LCBench.

Weaknesses:

1. The paper is not very well written that some parts of it is hard to read.

    1.1. Notation issues: 1) It should be $\alpha_\lambda, \beta_\lambda, \gamma_\lambda\in \mathbb{R}$ in eqn (3); 2) Why is $b$ bolded in eqn (4)(5)(6)(7)(8) but not bolded before and after?

    1.2. The paper is not self-contained that several techniques are used without introduction. For example, what is deep ensemble strategy? What is expected improvement? They seem to play an important role in the algorithm, but readers do not know what they are without referring to literature.

2. The empirical studies are puzzling.

    2.1. What is average normalized regret? Why is it an indicator of the efficacy of the HPO algorithms?

    2.2. I think a more convincing setting would be the following: 1) Choose a task with randomly initialized hyperparameter; 2) Apply DPL and other HPO methods; 3) Compare the final results (e.g. accuracy) of different methods; 4) Repeat these experiments to many tasks (NLP, CV etc.). After all, what we really care is the ultimate performance on the tasks.


**Summary Of The Paper:**

This paper proposes a hyperparameter optimization algorithm that leverages the power law phenomenon of the learning curves. In particular, the proposed method DPL assumes the surrogate $\hat f(\lambda)$  of the true learning curve $f(\lambda)$ obeys a power law function, which can be parameterized by a neural network. The ensemble of K diverse learned surrogates can be used to determine the next hyperparameter. Empirical studies show that 1) the power law assumption improves the quality of the learning curve prediction; 2) DPL achieves state-of-the-art results in HPO; 3) DPL explores the search space more efficiently than baselines.

**Summary Of The Review:**

The proposed method is novel in leveraging the power law in learning curves, but the clarity of the paper is a main concern. Besides, the evaluation is not convincing enough to show the efficacy of the proposed method.

---

> ### Author Response · Authors · 2022-11-18
> **Response to Reviewer 3aYj**
>
> Thank you for your thoughtful review and critical reading. Below we address your concerns:
>
> **Notation issues:**
>
> We thank the reviewer for his feedback. We have updated our submission accordingly following the given suggestions.
>
> **Submission not being self-contained:**
>
> We kindly refer the reviewer to Section 4, where we have described our method (Equation 6) and additionally the deep ensemble strategy. We agree with the reviewer that the definition of expected improvement is missing from the paper, and to make it self-contained we introduce it in Equation 7 as suggested.
>
> **What is the average normalized regret:**
>
> We kindly refer the reviewer to Section 6. In the updated Equation 9, where we describe the regret of a configuration, As an example, if the metric is accuracy (as in the case of LCBench, PD1), the higher the accuracy of a configuration the lower the regret, since the difference from the accuracy of the best (oracle) hyperparameter configuration is smaller.
>
> In Section 6, we additionally describe why we use the normalized regret to have an equal weighting from all datasets in the averaged results.
>
> Unfortunately, because the benchmarks are diverse, there is not one common metric for all. For LCBench and PD1 we use accuracy, while for TaskSet only loss is offered from the benchmark. We have updated Sections 6 and 6.1 to further improve the readability of our work.
>
> **I think a more convincing setting would be…:**
>
> We point the reviewer to Section 7 (Hypothesis 1, 2, 3), where, for every experiment shown in the paper, we run on different tasks (NLP, Tabular, Vision (CIFAR10, CIFAR100, ImageNet)) and every method starts with a single random hyperparameter configuration and follows an independent hyperparameter optimization process based on respective inner workings which is exactly what the reviewer suggests.
>
> **Reproducing the paper results:**
>
> We believe to have provided all the necessary information throughout our work so that a reader could replicate our method and experiments (We refer the reviewer to Section 6 and Appendix A). Furthermore, we would kindly point the reviewer to the abstract where we have provided a link to our open-sourced implementation. Lastly, in the code, we have provided the seeds used for our experiments.
>
> We believe to have answered and clarified all the concerns raised by the reviewer and we would kindly ask the reviewer to update the score. We are more than happy to answer any other questions if more should arise.

---

> > ### Comment · Reviewer_3aYj · 2022-12-08
> > **Thanks for the response**
> >
> > The authors has addressed some of my concerns. However, while the empirical studies seem to show that the proposed method is better than the competitors, the readability issue of the paper still exists. In particular, it is still not self-contained -- I still don't know what Deep Ensemble is without referring to another literature. Like one of the reviewers, I also suggest the authors spend some time polishing the paper, so that it is more readable and the superiority of the method is sufficiently highlighted. Given the current status of the draft, I will keep my score unchanged.

---

> > > ### Author Response · Authors · 2022-12-08
> > > **Response to Reviewer 3aYj**
> > >
> > > We thank the reviewer for the response and we provide the following clarification:
> > >
> > > - **Paper not being self-contained, deep ensemble not explained:**
> > >
> > >     We would kindly point the reviewer to Section 4, where we have described our method formulation and the deep ensemble
> > >     strategy. We have described how the uncertainty and the mean is generated by using an ensemble of $K$ surrogates in Equation 6, additionally, we have
> > >     clearly described the formulation of each surrogate in Equation 3 and 4. Furthermore, we quote the following from Section 4 of our work:
> > >     **"by initializing each surrogate with different weights and by training with a different sequence of mini-batches"**
> > >
> > > The description provided above **quoted from our paper** exactly describes deep ensembles and as such, we believe the paper is self-contained. **If the reviewer disagrees, we would like to kindly ask the reviewer to explicitly mention what is exactly missing from our work in understanding deep ensembles.**
> > >
> > > We would like to remind that the reviewer has rated our work with a **2** in terms of correctness, described as: **Several of the paper’s claims are incorrect or not well-supported.** and the empirical novelty with a **2**, described as: **The contributions are only marginally significant or novel**.
> > >
> > > Based on the reviewer's response after our initial clarification regarding the reproducibility and the suggested experiment: **"empirical studies show that the proposed method is better than the competitors"**, we believe that we have addressed both concerns in the score descriptions regarding the aforementioned score points.
> > >
> > > We believe to have clarified all the concerns and weaknesses raised by the reviewer and we would like to kindly invite the reviewer to update the score following the given clarifications. If the reviewer is not satisfied with our provided clarifications, we would kindly invite the reviewer to raise the concerns that were not addressed in detail, we would be happy to answer them.

---

> > > > ### Author Response · Authors · 2022-12-08
> > > > **Response to Reviewer 3aYj Part 2**
> > > >
> > > > Additional to the deep ensemble clarification, we provide the reviewer with the **changes introduced by our latest revision** prior to the start of the second discussion phase compared to the initial draft, where, **we refined our draft according to the feedback from all reviewers**: https://openreview.net/revisions/compare?id=NZ8Gb5GOrRu&left=aAhqrYUM3w&right=GL3lFCsOX&pdf=true
> > > >
> > > > As a summary of the changes, to improve the clarity of our work, we described in more detail:
> > > >
> > > > - The regret measure and the relation it has to the efficacy of the HPO algorithms.
> > > > - The metric used in all benchmarks.
> > > > - We incorporated EI by introducing equation 7 into the paper to make our work self-contained.
> > > > - We further clarified how the HPO methods are run in the experiments. In particular, we described that the methods start with a randomly sampled hyperparameter configuration.
> > > > - We introduced a more detailed description of all benchmarks and the hyperparameter search space they include.

---

### Author Response · Authors · 2022-11-19
**Summary of the main points**

We would like to thank all the reviewers for their thoughtful reviews and critical reading. Below we provide a summary of the main criticism points from all reviewers and our replies:

- **What is regret, could you provide plots for accuracy which is a more commonly used metric (Reviewers 3aYj, KEEA):**

    The regret is simply the distance from the performance metric of the oracle configuration of a dataset from the best-found configuration of every individual method during its optimization trajectory. We have additionally described why we normalize the regret so that the results clearly represent the performance on all datasets. The metrics offered by the benchmarks differ unfortunately, however, in the case of LCBench and Benchmark this is **accuracy**, while in the case of TaskSet this is **loss**.


- **It is unclear how the method would perform in a continuous search space (Reviewer KFvH):**

    We believe the question to be quite intriguing, however, we believe the issue is not specific to DPL, it pertains in general to multi-fidelity methods, where it is not certain how many times a configuration can be suggested between the different fidelity levels.


- **It is unclear how DPL would behave in the presence of learning rate schedules (Reviewer KFvH, vKZP):**

    Our motivational example features a step-wise cosine annealing learning rate scheduler. Additionally, TaskSet and PD1 offer various types of learning rate schedulers (exponential, polynomial, linear) whose behavior is additionally controlled by the search space configurations.

- **What is the performance of DPL on individual datasets, since, the average regret results might not represent the situation clearly (vKZP):**

    Figure 4 shows the statistical significance of the results during different stages of the optimization procedure and it additionally shows the rank of every method on a dataset, averaged for all datasets.

- **How fast is DPL, and what amount of training points does it need to offer high-quality predictions (Reviewer vKZP):**

    In Figure 5, we show the time performance of our method, which is normalized per dataset by the time it took random search to finish the optimization procedure, excluded after that point. We additionally emphasize that our method is not pretrained and it starts with 1 hyperparameter configuration, evaluated for 1 epoch (smallest fidelity). From that point, the surrogate takes over. Figure 4, showcases the performance as more BO iterations are performed and as our method has access to more points, yielding more qualitative predictions. The amount of iterations selected is up to the practitioner. We have provided detailed time information regarding our method in the respective answer.

We again thank all the reviewers for their feedback in helping us make our work better. We have modified the paper following all the reviewer's suggestions. For a more in detail answer to all the concerns, we refer the readers to the respective answers of the individual reviewers.

---

> ### Author Response · Authors · 2022-12-01
> **Summary of the second round of points:**
>
> We thank the reviewers for their additional questions and we provide a summary of the main points:
>
> - **Specific learning rate decay scenario and a request for including transformer architectures in the results (Reviewer KFvH):**
>
>     The reviewer asked for a scenario where the learning rate is kept nearly constant and is decayed drastically around 80% of the total
>     optimization, we provide evidence that the included scenario is part of our search space, results for which we have presented in Section
>     7. We also provide evidence that transformer architectures are already included in our results.
>
> - **Slow convergence for DPL in the pd1 benchmark (Reviewer vKZP):**
>
>     We provide new results showing better convergence of our method and better any-time results, outperforming all the considered
>     baselines and having the best rank compared to all baselines in the end.
>
> - **Missing PD1 and TaskSet plots for efficient exploring of the search space (Reviewer vKZP):**
>
>     We provide the missing plots showing that DPL does indeed explore the search space more efficiently in all benchmarks and
>     **furthermore proving that it is not only a pattern observable in a specific benchmark**.
>
> For a more in detail answer to all the concerns, we refer the readers to the respective answers of the individual reviewers.
> We welcome any new questions regarding our work.

---

### Decision · Program_Chairs · 2023-01-20

**Decision:**

Reject

**Justification For Why Not Higher Score:**

The authors did a very good job of answering questions and concerns from the reviewers. Some reviewers did adjust their assessment based on the interaction, however the majority of the reviewer did not find strength outweigh weakness and evaluated the paper as still below acceptance threshold for ICLR. Reviewers suggested significantly improving papers readability and highlighting superiority of PDL better would be ways to improve the paper but would require significant change.

Even with the author discussion and discussion among reviewers, no reviewer was in strong support for the paper acceptance.


**Justification For Why Not Lower Score:**

N/A

**Metareview: Summary, Strengths And Weaknesses:**

The paper proposes a new method of hyperparameter optimization (HPO) called Deep Power Law (DPL) utilizing scaling law behavior of loss curve being power-law. As scaling law is becoming a more and more important toolkit for understanding neural networks, the authors utilize the power-law property for hyperparameter optimization.

Authors demonstrate on 3 benchmarks (tabular, vision and NLP datasets) that DPL outperforms other 7 SoTA benchmarks.

The strength of the paper acknowledge by the reviewers are:
- Motivation is clear.
- Novelty of utilizing scaling law property of learning curve for HPO.
- Provides code and implementation details
- Empirical results are strong over 3 benchmarks on various domains

Here are few weakness raised by the reviewers:

- Limitation to continuous search space
- Use of learning rate schedules often leads to a non power-law training curve which is in tension the main assumption of the paper. While authors argued that benchmark suites do contain tasks with learning rate schedules, reviewers did not find that tasks included in the benchmark to be sufficient to ease the concern.
- While some reviewers did find the paper to be clear, others found issues with clarity and quality. They had a hard time finding details of the algorithm and implementation to the extent the main basis for rejection.